# Synthesis and Magnetic Properties of Carbon Doped and Reduced SrTiO₃ Nanoparticles

**Marina V. Makarova** [1,2,*], **Andrey Prokhorov** [1], **Alexander Stupakov** [1], **Jaromir Kopeček** [1], **Jan Drahokoupil** [1], **Vladimir Trepakov** [3] **and Alexander Dejneka** [1]

1   Institute of Physics AS CR, Na Slovance 2, 182 21 Prague 8, Czech Republic
2   Graduate School of Engineering Science, Akita University, 1-1 Tegatagakuenmachi, Akita 010-0852, Japan
3   Ioffe Physical-Technical Institute RAS, 194 021 St-Petersburg, Russia
*   Correspondence: makarova@fzu.cz

**Abstract:** We report on the studies of the synthesis, structural, and magnetic properties of undoped SrTiO₃ (STO), carbon-doped STO:C, and reduced STO STO:R nanoparticles. Fine (~20–30 nm) and coarse (~100 nm) nanoparticles with a single phase of cubic perovskite-type structure were sintered by thermal decomposition of SrTiO(C₂O₄)₂. Magnetization loops of fine STO:C and STO:R nanoparticles at low temperatures and an almost linear decrease in magnetization with temperature indicate the realization of a soft, ferromagnetic state in them, with a pronounced disorder effect characteristic of doped dilute magnetic semiconductors. Oxidation and particle size increase suppress the magnetic manifestations, demonstrating the importance of surface-related defects and oxygen deficiency in the emergence of magnetism. It was found that oxygen vacancies and doping with carbon make similar contributions to the magnetization, while complementary electron paramagnetic resonance, together with magnetization measurement studies, show that the most probable state of oxygen vacancies, which determine the appearance of magnetic properties, are charged F⁺ oxygen vacancies and C-impurity centers, which tend to segregate on the surface of nanoparticles.

**Keywords:** nanoparticles; strontium titanate; *p*-impurities and oxygen vacancies; magnetism

## 1. Introduction

The emergence of magnetic properties in non-magnetic materials with non-magnetic defects and impurities attracts particular academic and technological interest. Unfolded theoretical and experimental studies of bulk and nanoscale non-magnetic oxides (e.g., [1–22]) have found that the effective non-magnetic defects leading to magnetism are oxygen vacancies (V_O) [1,3,9,13–21], related complexes, nonstoichiometry [3,5,12,23], and non-magnetic *2p-* dopants (e.q. C, B, N or P) substituting for oxygen atoms (C@O) [4–6,10,11]. Moreover, it turns out that the appearance of magnetism is especially manifested and is a quite general property of nanosized objects with developed surfaces [13,14,24–31]. Particular interests of such researches are of highly polarized non-magnetic $d^0$ ABO₃ perovskite-type ferroelectrics and related oxides (e.g., [1,3,5–21,26–29,32,33]), opening up new possibilities for simultaneous control of the degree of freedom of charge and spin in these widely used functional materials of oxide electronics.

The main micromechanism of the occurrence of ferromagnetism caused by oxygen vacancies (Vo) is most often associated with the appearance of relatively deep spin levels in the band gap caused by vacancies. Such levels can mostly capture one electron, while the second electron occupies the conduction band. In ABO₃ perovskites, the presence of Vo promotes the appearance of Ta⁴⁺ and/or Ta³⁺ (Ti³⁺ and/or Ti²⁺) B- ions with a nonzero net spin [13,16,17,19]. An increase in the concentration of Vo and B- ions with nonzero spin on the surface and boundaries of grains and nanoparticles causes the formation of clusters, causing the possibility of a strong ferromagnetic exchange interaction.

The occurrence of magnetism caused by *sp*-elements doping, theoretically considered in [4–6,10,11,22], is much less investigated, especially experimentally. According to theory, magnetism induced by oxygen substitution with *p*-impurities occurs in the event that the orbital energies of their *p*-states are localized in the band gap of the host crystal, causing the possibility of spontaneous spin polarization, so the system gains magnetic semimetal or semiconducting properties [5,6]. The hole-mediated mechanisms of ferromagnetic interaction considered in [4] for ZnO with C@O carbon impurities should also be mentioned. Such substitution introduces O2*p* holes, which couple with the parent C2*p* localized spins by a *p*–*p* interaction, similar to *p*–*d* hybridization in semiconductors and oxides with TM impurities.

As far as we know, magnetism induced by *p*-impurities have been reported in a very limited number of experiments, in which carbon was used as an introduced impurity in C-implanted ZnO films and nanowires [4,30], $TiO_2$ ceramics [8], and in $TiO_2$/graphite carbon nanocomposites [31]. To the best of our knowledge, the possibility of the appearance of *p*-impurities induced magnetism in $d^0$ $ABO_3$ perovskite-like oxides has not been yet investigated experimentally.

In the present work, we report on the synthesis and study of the C-doped nanoparticles of STO, a comprehensively studied model archetype of the $ABO_3$ perovskite-type ferroelectric oxides, with a wide range of technological applications. It is worth noting that synthesis of such material, the observation and estimation of the magnetic contribution introduced by carbon doping in STO, are a challenge. Synthesis of C-, N- and S- doped $SrTiO_3$ has been reported for catalytic applications [34,35]. According to the DFT calculation [10], C impurity preferentially replaces Ti rather than Sr and O ions in STO, while spin-polarized states are formed only for C@O substitution; attainment of the concentration of C@O centers sufficient for the manifestation of magnetism is complicated. Moreover, such doping requires synthesis/treatment in reducing conditions ($H_2$, CO atmospheres, solid carbon additions etc.) to remove lattice oxygen and substitute it with carbon. Such a process is accompanied by a partial removal of lattice oxygen and the formation of oxygen vacancies [35–38]. Since vacancies of $V_O$ themselves lead to the magnetic moment emergence, the problem of identification, estimation, and comparison of $V_O$ not substituted by carbon and C@O impurity contributions to magnetism becomes a complex task. It is also necessary to eliminate the possible presence of accidental contaminating impurities of magnetic cations in the materials under study. We planned to try to develop technology to synthesize and study the properties of the STO nanopowders doped with carbon (STO:C)-exhibiting magnetic properties. Particular attention was paid to the identification, evaluation and comparison of the magnetic contributions coming from $V_O$ and C impurities. For this purpose, nanoparticles of various sizes, subjected to additional thermal treatment in a reducing and oxidizing atmosphere, were synthesized and studied too.

## 2. Materials and Methods

The method of complex compound thermolysis was used for nanoparticle fabrication. Strontium titanyloxalate was chosen as the basic precursor, with a Sr:Ti:C ratio of 1:1:4 [39] so the carbon dopant gets uniformly introduced simultaneously with the phase formation. An excess of carbon was taken to ensure maximal possible introduction of carbon into the lattice, which may depend on particle size and annealing temperature. Such a scheme was expected to decrease both reaction temperature and particle size. Figure 1 demonstrates the used synthesis scheme. $Ti(OCH(CH_3)_2)_4$ (Fluka, Buchs, Switzerland) was mixed with $H_2C_2O_4 \cdot 2H_2O$ (Lachema, Brno, Czech Republic) water solution in a 1:2 molar ratio. Concentrated $NH_4OH$ solution (Lachema, Brno, Czech Republic) was added until there was neutral reaction of the indicator paper to finalize titanyloxalate $TiO(C_2O_4)_2{}^{2-}$ formation. Then, $Sr(NO_3)_2$ (Sigma-Aldrich, St. Louis, MO, USA) solution was added to the titanyloxalate solution while stirring and then stirred for 30 min. The $Sr(NO_3)_2$ precursor was prior heated at 150 °C for 1 h to decompose possible hydrates. The white crystalline precipitate $SrTiO(C_2O_4)_2$ was collected, filtered, washed with distilled water, and dried.

Then, two parts were annealed in $H_2$/Ar flow (both Linde, Dublin, Ireland, 20 sccm each) for 1 h at 700 or 1000 °C, yielding carbon-doped STO:C (in fact, reduced carbon-doped) «fine» and «coarse» powders, marked as $C_f$ and $C_c$, respectively; we expected that the annealing at a higher temperature would result in a larger particle size. Another part was annealed at 700 °C in air for 1 h, to burn all the carbon dopant and obtain fine, undoped STO (marked as $I_f$) and then treated in the same reducing way as the initial precursor (see Figure 1, samples marked $R_f$ and $R_c$ for fine and coarse powders for low- and high-temperature annealing). Finally, fine C-doped powder was annealed in air for 1 h at 700 °C for oxidation (marked $OC_f$).

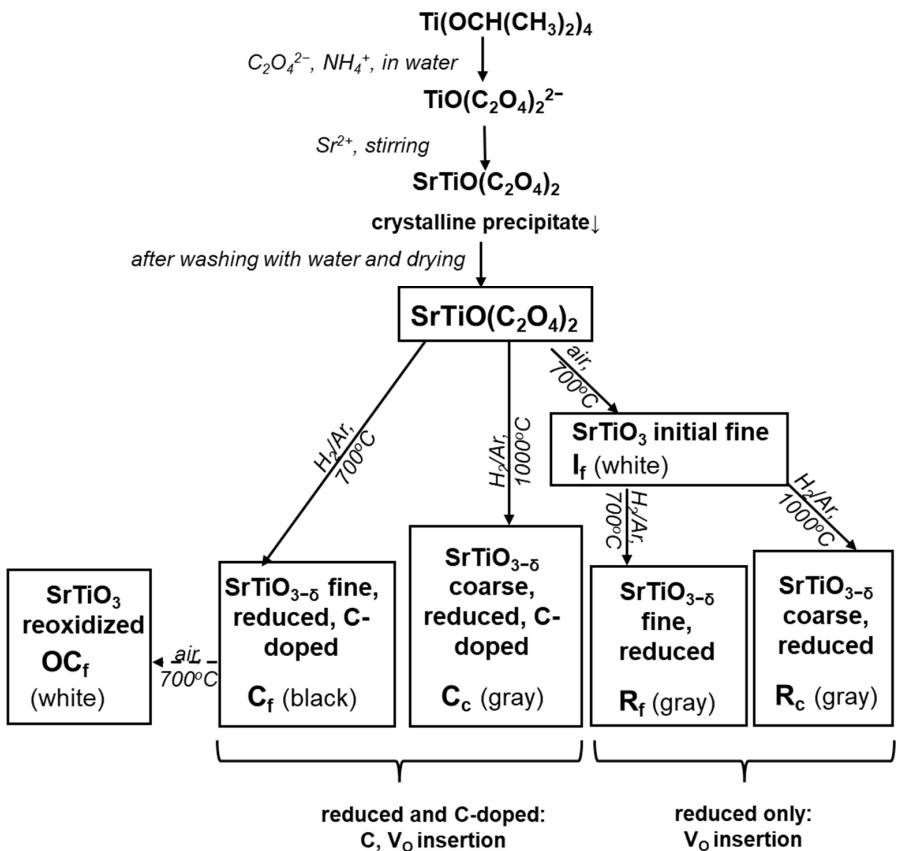

**Figure 1.** Scheme of the nanopowders synthesis. Powder colors are given for reference.

The resulting powder phase composition and structure were inspected by X-ray diffraction (XRD) using the PANalytical X'pert diffractometer (Malvern PANalytical, Malvern, United Kingdom) at $CoK_\alpha$ radiation, with a small addition of $CoK_\beta$. The obtained spectra were recalculated in $CuK_\alpha$ and compared against the standard STO spectra and the spectra of possible phase admixtures. The lattice constants were derived from the spectra fit in a PowderCell software 2.4. by W. Kraus and G. Nolze (Berlin, Germany) using the Ag addition to the studied powder as an internal standard. A scanning electron microscopy (SEM) experiment was performed with a TESCAN FERA 3 scanning electron microscope (TESCAN, Brno, Czech Republic), equipped with an OXFORD Instruments X-max EDS SDD 20 mm$^2$ detector (SEM-EDS) (Oxford Instruments, Abingdon, United Kingdom). To determine particle size distributions, 30+ individual nanoparticles were selected from SEM images and statistically analyzed. The same sets were used to calculate specific surfaces. Magnetic properties in the 10 K ÷ 300 K temperature region were characterized by the SQUID Magnetic Property Measurement System (Quantum Design) MPMS-XL 7T (San Diego, CA, USA).

The absence of accidental contamination of experimental samples with magnetic ions, such as Fe, Mn, etc., which could be a source of magnetism, was controlled by the electron

paramagnetic resonance (EPR) method. Along this way, we also hoped to get data on the centers involved in the formation of the observed occurrence of actual ferromagnetism. Continuous wave EPR spectra were measured using a Bruker X/Q band E580 FT/CW ELEXSYS spectrometer (Bruker, Billerica, MA, USA) in the temperature range of 5–300 K. An ER 4122 SHQE Super X High-Q cavity with TE011mode was used. The samples were placed into quartz tubes with a diameter of 4 mm. The experimental parameters were: microwave frequency $9.8756 \pm 0.0005$ GHz, microwave power 1.500 mW, modulation frequency 100 kHz, modulation amplitude 0.2 mT, and the conversion time 60 ms. Neither magnetic admixture lines nor ferromagnetism were found in the starting materials.

### 3. Results and Discussion

Figure 2 shows the XRD spectra of the reference fine STO ($\mathbf{I_f}$), reduced undoped ($\mathbf{R_f}$ and $\mathbf{R_c}$), and reduced C-doped ($\mathbf{C_f}$ and $\mathbf{C_c}$) nanopowders and standard peaks of STO marked with black bars. The designations and preparation details are given in the Figure 1 scheme and Table 1. A comparison of the standard XRD peaks of STO indicates that all the powders have a cubic perovskite-type structure, with some traces of $SrCO_3$ admixture in $\mathbf{I_f}$ and $\mathbf{R_f}$ powders, which might appear due to slight Sr excess in the initial precursor.

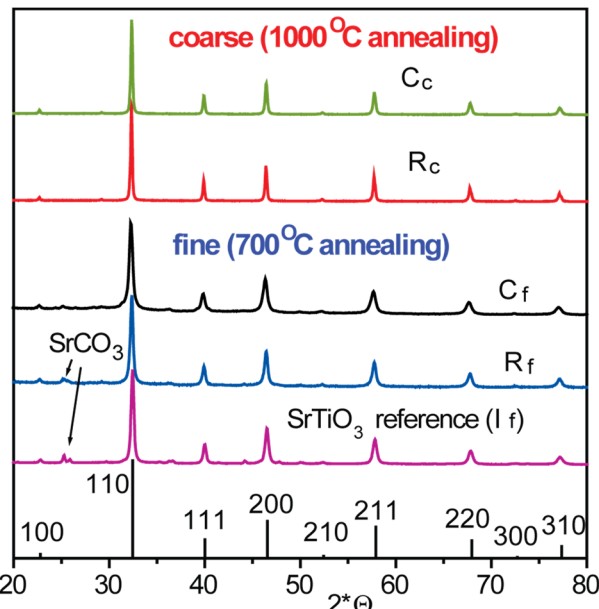

**Figure 2.** XRD spectra of the fine and coarse nanopowders with ($\mathbf{C_f}$, $\mathbf{C_c}$) and without carbon doping ($\mathbf{R_f}$, $\mathbf{R_c}$). Reference ($\mathbf{I_f}$) STO powder curve and standard peaks of STO (black bars) are given for comparison.

Figure 3a presents an SEM image of a typical $\mathbf{C_f}$ powder. As it can be seen, the particles have a more or less spherical shape and are aggregated into flower-like structures, possibly inherited from the basic $SrTiO(C_2O_4)_2$ crystalline precursor. Other fine powders have a morphology similar to that of $\mathbf{C_f}$, as demonstrated in a Figure 3b–d. Particle size distributions are shown in the Figure 3e.

Particle size was roughly the same for all the powders obtained at 700 °C and slightly increased upon oxidization in $\mathbf{C_f}$ samples ($\mathbf{OC_f}$). Powders prepared at 1000 °C were coarser and had a broader particle size distribution due to more intensive crystallite growth. Thus, powders denoted as "f" or "fine" have ~20–30 nm particle size while "coarse" are ~100 nm. Table 2 shows the lattice constants and their relationship with particle size. It is seen that they do not largely differ between each other. However, slight lattice expansion was observed for the carbon-doped $\mathbf{C_f}$ particles, which is consistent with the data for Ti oxycarbide [40–42] and can be related to partial substitution of O ions with C (formation of C@O centers). In the oxidized C-doped powder $\mathbf{OC_f}$, the lattice constant did not

undergo visible changes, probably because the oxidation temperature was insufficient for full decarbonization and affected only near-surface layers. Coarse powders, annealed at 1000 °C, have a lower lattice constant, compared to 25–30 nm powders, due to the effect of unit cell expansion in the fine powders [43,44]. The carbon-doping effect was not seen on them, probably because oxycarbide formation depends on particle size as well [36].

**Table 1.** Sample description and notations.

| Powder Descriptions | Designation |
|---|---|
| SrTiO$_3$ initial<br>SrTiO(C$_2$O$_4$)$_2$ precursor annealed at 700 °C in air | **I$_f$** |
| SrTiO$_3$ fine reduced,<br>**I$_f$** annealed at 700 °C in H$_2$/Ar flow, | **R$_f$** |
| SrTiO$_3$ coarse reduced<br>**I$_f$** annealed at 1000 °C in H$_2$/Ar flow | **R$_c$** |
| SrTiO$_3$ fine C-doped<br>SrTiO(C$_2$O$_4$)$_2$ precursor annealed at 700 °C in H$_2$/Ar flow | **C$_f$** |
| SrTiO$_3$ white<br>**C$_f$** oxidized at 700 °C in air | **OC$_f$** |
| SrTiO$_3$ coarse C-doped<br>SrTiO(C$_2$O$_4$)$_2$ precursor annealed at 1000 °C in H$_2$/Ar flow | **C$_c$** |

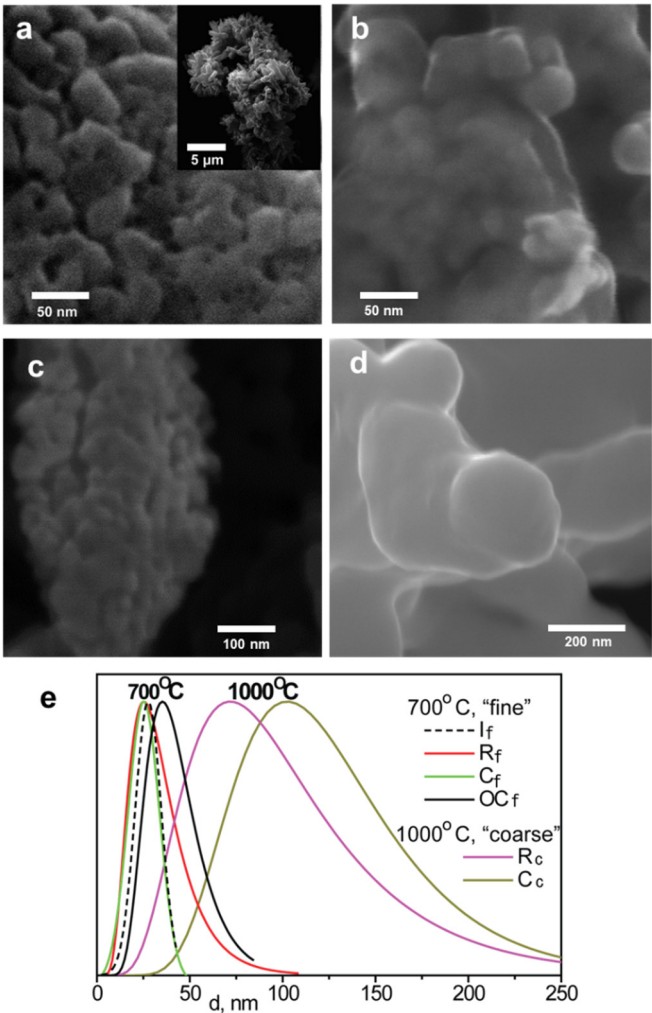

**Figure 3.** SEM image of (**a**) **C$_f$** nanoparticles and their as-grown aggregates (inset), (**b**) **R$_f$**, (**c**) **OC$_f$**, and (**d**) **C$_c$** nanoparticles. (**e**) Particle size distributions for fine and coarse nanopowders.

**Table 2.** Lattice constant *a* and particle size derived from XRD and SEM data.

| Designation | $a$, Å | Particle Size, nm |
|:---:|:---:|:---:|
| $I_f$ | 3.909 | 28 |
| $R_f$ | 3.910 | 25 |
| $C_f$ | 3.913 | 25 |
| $OC_f$ | 3.913 | 40 |
| $C_c$ | 3.907 | 100 |
| $R_c$ | 3.907 | 70 |

Figure 4 presents the results of magnetic studies for C-doped ($C_f$, $C_c$) and undoped ($R_f$ and $R_c$) powders with fine (~20–30 nm) and coarse (~100 nm) particle size: (*a*) M–H hysteresis loops at T = 10 K and for C-doped at 300 K (Inset), (*b*) temperature dependence of magnetization for reduced and C-doped powders, and (*c*) saturation magnetization on nanopowder specific surface area at 10 K. It is seen that both coarse $R_c$ and $C_c$ nanopowders hysteresis loops are not unfolded, characterizing a slight nonlinearity of the M–H field dependence. At the same time, the M–H behavior of fine $C_f$ and reduced $R_f$ nanopowders clearly show unsaturated narrow hysteresis loops, which are very close to those observed in $BaTiO_3$, $SrTiO_3$, and $KTaO_3$ crystals implanted with Co and Mn [45], exhibiting the properties of a soft ferromagnet. In Figure 4b, it is also seen that the magnetization of fine $R_f$ and $C_f$ is greater than that of the others, and, as in [45], almost linearly decreased with temperature, in contrast to the usual Brillouin temperature dependence that is most likely attributed to the disorder inherent in dilute magnetic semiconductors [46]. Similar behavior is observed for pure $I_f$ and oxidized $OC_f$ fine powders, in which the observed magnetization is much less pronounced than for reduced $R_f$ and $C_f$ powders; hysteresis loops are very narrow and not unfolded and more meet the nonlinear M–H dependence. Figure 4a–c shows that the highest magnetization value is achieved in $C_f$ and $R_f$ nanoparticles with the minimum particle size and a large specific surface area, which indicates the importance of near-surface defects in the magnetism emergence [19–21]. Reasonably assuming that a certain fraction of oxygen vacancies in $C_f$ particles is not replaced by carbon, we conclude that doping with both C and Vo induces close magnetization values and is efficient in the emergence of ferromagnetism.

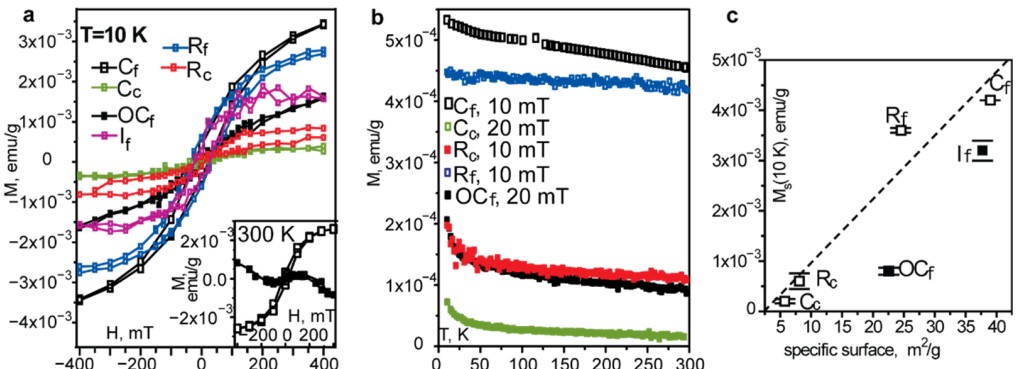

**Figure 4.** (**a**) Magnetization loops at T = 10 K and at 300 K (Insert), (**b**) temperature dependence of the magnetization, and (**c**) saturation magnetization on specific surface area at 10 K for reduced (open squares) and oxidized (solid squares) powders. Lines are only guides for the eye.

Figure 4c demonstrates the increasing dependence of magnetization on specific surface areas in both reduced and oxidized powders, which may be related to the contribution of natural and artificial surface defects, oxygen vacancies, and dangling bonds. Ferromagnetism is almost suppressed in coarse $R_c$ and $C_c$ powders with low surface area, despite their thermal reduction and C doping. However, magnetization is pronounced in the fine nanopowders with large surface $C_f$, $R_f$, and $I_f$, where reduction and C doping additionally increases it in $R_f$ and $C_f$, compared to reference $I_f$.

The M–H curve of C-doped nanopowder after thermal oxidation (**OC$_f$**) significantly differs from the initial **C$_f$** powder at both 10 and 300 K (Figure 4a,b). Ferromagnetism almost disappears, while the particle size (Figure 3e) and surface area (Figure 4c) do not change much. After oxidation, the **C$_f$** powder changes color from black to white, which may indicate both carbon dopant elimination and V$_O$ filling in the resulting **OC$_f$** sample. Taking into account that the EPR data presented below did not reveal any noticeable manifestations of the presence of Fe, Mn ions, and other magnetic impurities, which often contaminate perovskite-like oxides, we conclude that ferromagnetism in **C$_f$** and **R$_f$** is mainly due to "non-magnetic doping," with carbon and with oxygen vacancies localized mainly in the near-surface regions of the nanoparticles. The resulting magnetic state is most likely the state of a soft ferromagnet, with the pronounced effect of the disorder characteristic of doped dilute magnetic semiconductors.

It should be noted that, concluding about the realization of a soft ferromagnetic state, it should be taken into account that ferrimagnets can also exhibit magnetic properties similar to those we observed. However, unlike ferrimagnets, STiO$_3$ has a very simple structure in which the presence of various magnetic structural complexes in various crystalline environments leading to ferromagnetism seems extremely unlikely and has never been observed in SrTiO$_3$ doped with magnetic impurities; it often exhibits the properties of dilute magnetic semiconductors (e.g., [45]). We do not know of any publications reporting the formation of a ferrimagnetic state induced by nonmagnetic defects and impurities, not only in strontium titanate and related ABO$_3$ oxides, but also in oxides of simple crystal structure. Finally, from Figure 4a, it is seen that the hysteresis loops, the values of magnetization **C$_f$** and, especially important, **R$_f$** nanopowders with one type of defects (Vo) leading to magnetism are very close, which complicates the ferrimagnetic interpretation and allows us to assume a soft ferromagnetic state.

Figure 5 shows the broadband (Figure 5a) and short-range (Figure 5b) EPR spectra of the nanopowders under a study recorded at room temperature, and the short-range spectra for T = 25 K (Figure 5c,d). For all the Figures, the mass of the samples was fixed as 20 mg, with the exception of Figure 5a. It is seen that the EPR spectra at room temperature contain two wide bands that are at the noise level and will not be discussed, along with prominent pronounced absorption lines in the region of ~335 mT ($g \approx 2.002$ $A$-lines). Figure 5b shows the detailed structure of $A$-lines, as well as the presence of another weak resonance in the region of $g \approx 1.978$. Figure 5c presents the EPR spectra at T = 25, of which the most intense are the $A$-lines (Lorentzian shape with a linewidth of ~4 mT) of **C$_f$** nanopowder, and the weaker but quite pronounced are the $A$-lines of **C$_c$** and **R$_f$**.

The position of the EPR absorption $A$-lines in the region of $g \approx 2.002$ meets the condition of resonance absorption with the spin s = $\frac{1}{2}$, which can refer to a number of centers, such as itinerant electrons, which provide magnetic ordering [7,47,48], electrons in the conduction band and electrons of spin levels inside the energy gap, originating from the valence band upon doping with Vo, F-centers on the surface of nanoparticles, $p$ impurities, and C dopants [4,5,14,31,49–52].

It is reasonable to assume that the $A$-lines of the EPR resonance of carbon-free fine reduced **R$_f$** nanopowders with the largest surface area are associated with oxygen vacancies, which, according to [9,14,15,20,53–56], are predominantly charged F$^+$ centers, with a fairly deep localization, low formation energy, and a greater tendency to segregate on the STO surface than neutral triplet vacancy F-centers (with two electrons). The significant role of oxygen vacancies, especially surface ones, is convincingly confirmed by a strong decrease in the EPR signal and magnetization of carbon-free coarse **R$_c$** and a complete suppression of the $A$-EPR line of oxidized **C$_f$** particles, where thermal oxidation eliminates both oxygen vacancies and carbon dopants.

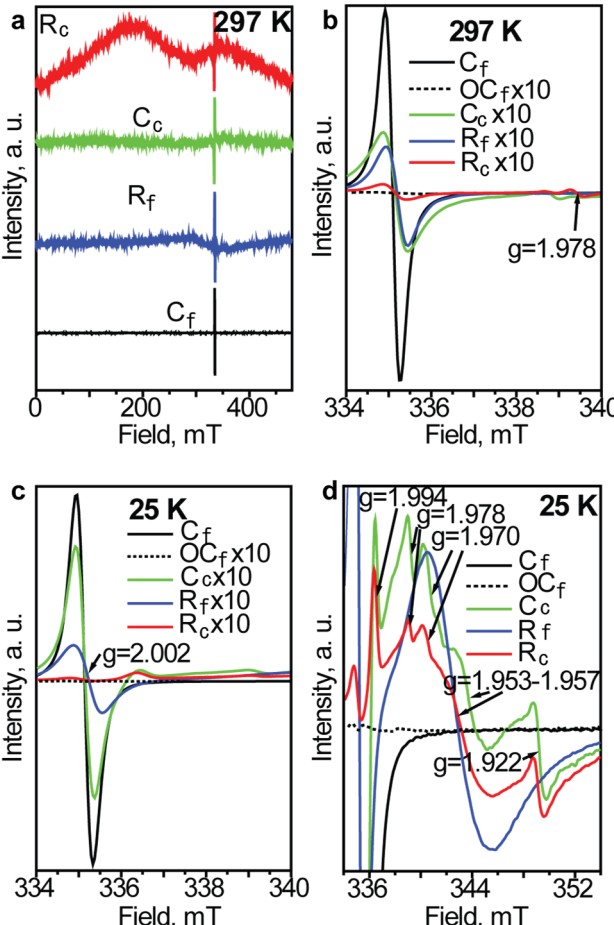

**Figure 5.** (**a**) Broadband and (**b–d**) short range EPR spectra of C-doped reduced (**C$_f$** and **C$_c$**) and undoped reduced (**R$_f$** and **R$_c$**) STO nanopowders taken at room temperature (**a,b**) and at 25 K (**c,d**). The spectrum of fine oxidized carbon-doped powder (**OC$_f$**) is given for comparison. For all the figures, sample mass was fixed as 20 mg, with the exception of Figure 5a.

The most interesting feature of the EPR spectra in the region g ≈ 2.002 is the absorption of **C$_f$** nanoparticles, which turns out to be much stronger than that of other nanoparticles and is slightly shifted towards larger values of the *g*-factor, compared to the spectrum of **R$_f$** nanoparticles associated with oxygen vacancies, which agrees well with the results of the EPR studies of C-doped titanium dioxide [52]. This makes it possible to quite confidently relate the EPR spectrum of **C$_f$** nanoparticles in the region of *g* ≈ 2.002 with the carbon impurity centers. Another source of free electrons and the appearance of *A*– lines in the EPR spectra could be the phases formed by free residual carbon. However, the presence of similar ones in our case should be negligibly small, since they are not visible in the XRD spectra, even in the form of halos. In addition, the soft ferromagnetism inherent in the main allotropic modifications of carbon (diamond, graphite, nanographite, nanotubes, fullerenes) [57], if present, is negligible, much weaker than that observed in our experiments.

Thus, the results obtained provide sufficient grounds to believe that the observed narrow EPR lines in the *g* ≈ 2.002 region mainly originated due to oxygen vacancy- and carbon-containing defect complexes.

Table 3 arranges the room temperature magnetization values from Figure 4a,b and the intensities of the EPR *A*-lines (Figure 5b) in decreasing order.

**Table 3.** Ranking from larger to lower magnitudes of room temperature magnetization (Figure 4a,b) and EPR *A*-line intensity (Figure 5b) in the powders under study.

| Magnetization | *A*–EPR Line Intensity |
| --- | --- |
| $C_f$ | $C_f$ |
| $R_f$ | $C_c$ |
| $OC_f$ | $R_f$ |
| $R_c$ | $R_c$ |
| $C_c$ | - |

It follows from Figures 4a and 5c and Table 3 that the most pronounced ferromagnetism at room temperature and the intensity of the EPR *A*-line were observed in carbon-doped $C_f$ and reduced $R_f$ "fine" STO powders with the most developed surface. Note that the smallest value of magnetization was obtained in $C_c$ nanopowders, while $C_f$ nanopowders with the largest surface area revealed the highest magnetization value and the intensity of the EPR *A*-line spectra. That allows it to assume the leading role in the production of the magnetism of C-related defect as it enters segregating in the surface regions.

Correlation was also observed in carbon-free $R_f$ and $R_c$ powders, where the *A*-line could be certainly associated with $V_O$. Both *A*-line intensity and magnetization were much higher for fine $R_f$, which can again testify that $V_O$ are preferably formed near the particle surface and are responsible for magnetic order emergence. This line is also absent in the reoxidized sample $OC_f$ after C and oxygen vacancies elimination, when magnetization disappears as well. These observations confirm the importance of the contribution of oxygen vacancies $V_O$, most likely associated with charged $F^+$ centers mentioned above, which have a low formation energy and a great tendency to segregate in the STO surface in magnetic order emergence.

Figure 5d shows the detailed resolved EPR spectra of a series of five weak resonance lines in the region of 334–354 mT (g ≈ 1.97) at 25 K, whose weak manifestation is noticeable already at room temperature (Figure 5b). With the highest probability, the found five resonance lines with $g_1 = 1.994$, $g_2 = 1.978$, $g_3 = 1.970$, $g_4 = 1.953$–1.957, and $g_5 = 1.922$ correspond to the responses of different types of Ti centers. $Ti^{3+}$ ions can be located as octahedral site ions in the distorted oxygen surrounding [58], in octahedral sites with two oxygen vacancies, and at Sr sites [59], and then the value $g_1$ can be associated with $Ti^{3+}$ adjacent to oxygen vacancies, with $g_x = 1.992$ and $g_z = 1.994$ obtained in the EPR spectra of reduced STO single crystals [59]. At the same time $g_2$ and $g_3$ lines can possibly be associated with $Cr^{3+}$ ions, replacing $Ti^{4+}$ (1.977, 1.974 [60–62], 9.780 [63]), an unavoidable impurity contaminating STO [64]. Note that the spectra of coarse $C_c$ and $R_c$ powders (Figure 5d) show individual EPR lines of $Ti^{3+}$ EPR lines, while on fine $R_f$, these lines are blurred. This, again, may be explained by the larger role of the surface in $R_f$ powder and by the reduction of coordination symmetry, thereby demonstrating again the connection between the occurrence of magnetic order and surface oxygen vacancies.

With regards to the properties of defects in the triplet (S = 1) state, they are studied scarcely, and their atomic nature is still tentative (see Refs. [65,66]). In this connection, the magnetic properties can be provided by triplet (S = 1) neutral oxygen vacancies ($F^0$), where two compensating electrons of similar spin are localized at neighboring equivalent $t_{2g}$ states of neighboring $Ti^{3+}$ ions. At the same time, the model was theoretically considered for STO [53–56] and experimentally for $TiO_2$ [66], and could manifest itself in the EPR spectra of STO. However, the interpretation of EPR lines, based on the comparison with the *g*-factor values of rutile [65], has its limitations and requires caution. According to calculations, ionization energy of the triplet neutral $F^0$ center in STO is very low (0.5 eV in the bulk and 0.2 eV on the surface). Therefore, the electrons of oxygen vacancy are weakly bound at Ti and most likely singlet. Due to low energy, $F^0$ centers can be ionized into charged vacancies, such as $F^+$ (with one electron $V^+_O$), having nonzero spin. This leads to the fact that the triplet oxygen vacancies turn out to be unstable, in contrast to the dominant $F^+$.

Note that the relative weakness of the EPR spectra associated with Ti ions clearly indicates the leading role of Vo and C-related defects in the production of the observed magnetism.

## 4. Conclusions

We developed a novel technology of carbon introduction into nanosized STO and studied the structure, magnetization, and EPR spectra of carbon-doped STO:C; we also reduced STO (STO:R) nanoparticles exhibiting the properties of a soft ferromagnet state with a pronounced disorder effect characteristic of doped dilute magnetic semiconductors, with ferromagnetism observed up to room temperature.

Particular attention was paid to the identification, evaluation, and comparison of the magnetic contributions coming from $V_O$ and C impurities. Despite the difficulty in estimating the concentration of defects in nanopowders doped with C, we concluded that the contribution to magnetism due to doping with C and $V_O$ was of the same order. It is important to note that the largest contribution to magnetism was made by C- and Vo- related defects segregating near the surface, which was accompanied by an increase in the magnetization of nanoparticles, with a decrease in their size and an increase in their surface area. Complementary electronic paramagnetic resonance and magnetization measurements made it possible to determine the EPR spectra connected to C impurities and charged oxygen vacancies $F^+$ located in the region of $g \approx 2.002$, which determine magnetic properties.

**Author Contributions:** Conceptualization, V.T.; methodology, V.T. and M.V.M.; software, A.P.; validation, V.T. and M.V.M.; formal analysis, M.V.M.; investigation, M.V.M., J.K., A.S., A.P. and J.D.; resources, A.S., J.D. and A.D.; data curation, M.V.M. and A.P.; writing—original draft preparation, M.V.M.; writing—review and editing, V.T.; visualization, M.V.M.; supervision, V.T. and A.D.; project administration, V.T. and A.D.; funding acquisition, A.D. All authors have read and agreed to the published version of the manuscript.

**Funding:** The EPR measurements were carried out within the SAFMAT infrastructure at FZU CAS. A. S. appreciates the Czech Science Foundation [grant number 20-21864S]. Magnetic properties were measured in the Materials Growth and Measurement Laboratory (www.mgml.eu, accessed on 3 September 2022), which is supported within the program of Czech Research Infrastructures [project number LM2018096]. CzechNanoLab project LM2018110, funded by MEYS CR, is gratefully acknowledged for the financial support of the SEM measurements. This work was supported in part by the Operational Program Research, Development, and Education financed by the European Structural and Investment Funds and the Czech Ministry of Education and Youth and Sports Project SOLID21 CZ.02.1.01/0.0/0.0/16_019/0000760.

**Institutional Review Board Statement:** Not applicable.

**Informed Consent Statement:** Not applicable.

**Data Availability Statement:** Not applicable.

**Acknowledgments:** V.T. kindly thanks E. Kotomin and R. Yusupov for valuable discussions. M.V.M. thanks F. Borodavka, A. Lynnik, and V. Gaertnerova for their help in conducting and processing experimental results.

**Conflicts of Interest:** The authors declare no conflict of interest. The funders had no role in the design of the study; in the collection, analyses, or interpretation of data; in the writing of the manuscript; or in the decision to publish the results.

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
