# Peer review of "Synthesis and Magnetic Properties of Carbon Doped and Reduced SrTiO3 Nanoparticles"

_crystals, doi:10.3390/cryst12091275_

Round 1
Reviewer 1 Report
Reviewer report Manuscript ID crystals-1892282
Dear Editor,
Current manuscript deals with the physical properties of undoped SrTiO3 10 (STO), carbon-doped, STO:C, and reduced STO, STO:R, nanoparticles. This paper is well organized and written clearly. This paper is perfectly suitable for the current journal. However, I suggest some minor changes and suggestions before the acceptance.
1) In the figure-4(a) why the magnetization of fine powders is grater than the coarse powders explain.
2) Provide the MH loop of pure SrTiO3 and confirm whether it is magnetic or non-magnetic in the manuscript.
3) Authors should provide the details that how they confirmed the compounds exhibits Ferromagnetic nature compared to Ferrimagnetic in the manuscript.
4) Line – 194, Authors should clearly explain how the Ferromagnetism disappears in figure-4(c) with the particle size and surface area.
5) Line-256, provide the correct compound “reduced carbon doped Cf ’’.
6) Regarding Figure 4c: to connect a straight line minimum three points are required (however, OCf and 1f are the only two points available). Include the error bars in this plot.
7) Include the Miller Indices to Figure 2 of XRD spectra.
8) What is the erroer/least count of the ESR frequency used in the present study (9.8756 GHz ± XXX)
Author Response
We thank reviewers for valuable and positive reports, detailed consideration and important remarks about possible improvements of our paper. We are happy that the referees like the manuscript and recommend its publishing in minor revised form. We have added some explanations (tracked) into the text and partially restructured the manuscript.
We hope that the current version would fit the Crystals standards to be published.
Here are the detailed responses:
REVIEWER 1
Reviewer 1: In the figure-4(a) why the magnetization of fine powders is grater than the coarse powders explain.
Authors: We added the explanation to Line 183 (now Line 187): “Figure 4a–c shows that the highest magnetization value is achieved in Cf and Rf nanoparticles with the minimum particle size and large specific surface area, which indicates the importance of near-surface defects in the magnetism emergence [19–21] .”
Reviewer 1: Provide the MH loop of pure SrTiO3 and confirm whether it is magnetic or non-magnetic in the manuscript.
Authors: Actually, oxidized OCf sample could be taken as pure STO. But for experimental purity we also added the MH loop of If sample (standard undoped). Some magnetism is present there due to surface defect states and broken bonds, as it has been reported [20,21]. However, magnetization values are lower than reduced or C-doped STO powders. We wrote in Lines 203-205:
“However, magnetization is pronounced in the fine nanopowders with large surface Cf, Rf and If, where reduction and C doping additionally increase it in Rf and Cf compared to reference If.”
Reviewer 1: Authors should provide the details that how they confirmed the compounds exhibits Ferromagnetic nature compared to Ferrimagnetic in the manuscript.
Authors: We thank referee for very interesting and important remark.
Indeed, the properties of a soft ferromagnet and a ferrimagnet are very similar, and the ultimate choice based on the data we have obtained within the framework of the problems being solved in this work is very difficult, if possible at all.
However, unlike ferrimagnets STiO3 has very simple structure in which the presence of various magnetic structural complexes in various crystalline environments leading to ferrimagnetism seems very unlikely. Ferrimagnet (non-singulent antiferromagnetic) ground state has never been observed in ABO3 relative perovskite like oxides as well as in SrTiO3 doped with magnetic impurities, which exhibits the properties of a dilute magnetic semiconductors (see, e.q.Ref. [45]).
Also, the formation of a Ferrimagnetic state was not reported in any of the works cited by us, in which the magnetism induced by oxygen vacancies, non-stoichiometry and p- impurities in strontium titanate and in oxides of a simple structure was studied.
At last, from Fig. 4a it is seen that the hysteresis loops and the values of magnetization Cf and, which is especially important, Rf nanopowders with one type of defects (Vo) leading to magnetism are very close, which complicates the ferrimagnetic interpretation and allows us to assume the state of a soft ferromagnet one.
As a result, taking into account the valuable, important remark of the referee, in the end of Results and Discussion section (Line 218) we added following:
“It should be noted that concluding about the realization of a soft ferromagnetic state, it should be taken into account that ferrimagnets can also exhibit magnetic properties similar to those observed by us. However, unlike ferrimagnets, STiO3 has very simple structure in which the presence of various magnetic structural complexes in various crystalline environments leading to ferrimagnetism seems extremely unlikely and has never been observed in SrTiO3 doped with magnetic impurities, which often exhibits the properties of dilute magnetic semiconductors (e.g. [45]). We do not know publications reporting the formation of a ferrimagnetic state induced by nonmagnetic defects and impurities not only in strontium titanate and related ABO3 oxides, but also in oxides of simple crystal structure too. At last, from Figure 4a it is seen that the hysteresis loops and the values of magnetization Cf and, which is especially important, Rf nanopowders with one type of defects (Vo) leading to magnetism are very close, which complicates the ferrimagnetic interpretation and allows us to assume the state of a soft ferromagnet one.”
Reviewer 1: Line – 194, Authors should clearly explain how the Ferromagnetism disappears in figure-4(c) with the particle size and surface area.
Authors: We added to Line 187 that Figure 4a–c shows that the highest magnetization value is achieved in Cf and Rf nanoparticles with the minimum particle size and large specific surface area, which indicates the importance of near-surface defects in the magnetism emergence [19–21].
We additionally added a clarifying paragraph about particle size effect in Figure 4c, Line 199:
“Figure 4c demonstrates the increasing dependence of magnetization on specific surface area in both reduced and oxidized powders, which may be related to the contribution of natural and artificial surface defects, oxygen vacancies and dangling bonds. Ferromagnetism is almost suppressed in coarse Rc and Cc powders with low surface area, despite their thermal reduction and C doping. However, magnetization is pronounced in the fine nanopowders with large surface Cf, Rf and If, where reduction and C doping additionally increase it in Rf and Cf compared to reference If.”
Reviewer 1: Line-256, provide the correct compound “reduced carbon doped Cf ’’.
Authors: Thank you for important remark! To avoid confusion we removed “reduced” in “reduced carbon-doped Cf” and added “STO” abbreviation. Now there is a Line 283.
Reviewer 1: Regarding Figure 4c: to connect a straight line minimum three points are required (however, OCf and 1f are the only two points available). Include the error bars in this plot.
Authors: We removed connecting line and calculated error bars.
Reviewer 1: Include the Miller Indices to Figure 2 of XRD spectra.
Authors: We included them.
Reviewer 1: What is the erroer/least count of the ESR frequency used in the present study (9.8756 GHz ± XXX)
Authors: 9,9.8756 GHz ± 0.0005 GHz. We added to the Line 131-132.
In conclusion, we hope that we have considered all the remarks and suggestions of the referees and believe that the manuscript now meets all requirements for publication in Crystals.
Reviewer 2 Report
-In page 2, line 71. The authors need to describe more about "reducing conditions".
-Figure 2. The standard XRD lines are too weak to be observed.
-The authors are suggested to also include SEM images of other samples in figure 3.
-How do you determine the particle size distributions?
-Apart from the SrTiO3, do other non-oxide type perovskite materials also exhibit magnetic properties? Are they share the same mechanism?
-The grammer should be improved.
Author Response
We thank reviewers for valuable and positive reports, detailed consideration and important remarks about possible improvements of our paper. We are happy that the referees like the manuscript and recommend its publishing in minor revised form. We have added some explanations (tracked) into the text and partially restructured the manuscript.
We hope that the current version would fit the Crystals standards to be published.
Here are the detailed responses:
REVIEWER 2
Reviewer 2: In page 2, line 71. The authors need to describe more about "reducing conditions".
Authors: For clarification we changed the sentence in Line 73 “Moreover, such doping requires synthesis/treatment in reducing conditions (H2, CO atmospheres, solid carbon additions etc.) to remove lattice oxygen and substitute it with carbon.”
Reviewer 2: Figure 2. The standard XRD lines are too weak to be observed.
Authors: Thank you for remark. We rearranged the figure 2 to make them more observable.
Reviewer 2: The authors are suggested to also include SEM images of other samples in figure 3.
Authors: We added representative SEM images of reduced, reoxidized and coarse nano-powders. The particle shapes are more or less the same, but the size may differ.
Reviewer 2: How do you determine the particle size distributions?
Authors: Thank you for good question. We measured particle sizes of nanoparticles in SEM images.
We added clarification to Materials and Methods, line 117:
“To determine particle size distributions, 30+ individual nanoparticles were selected from SEM images and statistically analyzed. The same sets were used to calculate specific surface.”
Reviewer 2: Apart from the SrTiO3, do other non-oxide type perovskite materials also exhibit magnetic properties? Are they share the same mechanism?
Authors: Such works are unknown to us, at least what concerns perovskites without magnetic elements.
Reviewer 2: The grammer should be improved.
We tried to correct the most evident errors, however, revision time was limited.
In conclusion, we hope that we have considered all the remarks and suggestions of the referees and believe that the manuscript now meets all requirements for publication in Crystals.